# Principal Component Analysis and Machine Learning Approaches for Photovoltaic Power Prediction: A Comparative Study

**Souhaila Chahboun *** and **Mohamed Maaroufi**

Mohammadia School of Engineers, Mohammed V University in Rabat, Rabat 10090, Morocco; maaroufi@emi.ac.ma
* Correspondence: souhaila_chahboun@um5.ac.ma; Tel.: +212-639507337

**Abstract:** Nowadays, in the context of the industrial revolution 4.0, considerable volumes of data are being generated continuously from intelligent sensors and connected objects. The proper understanding and use of these amounts of data are crucial levers of performance and innovation. Machine learning is the technology that allows the full potential of big datasets to be exploited. As a branch of artificial intelligence, it enables us to discover patterns and make predictions from data based on statistics, data mining, and predictive analysis. The key goal of this study was to use machine learning approaches to forecast the hourly power produced by photovoltaic panels. A comparison analysis of various predictive models including elastic net, support vector regression, random forest, and Bayesian regularized neural networks was carried out to identify the models providing the best predicting results. The principal components analysis used to reduce the dimensionality of the input data revealed six main factor components that could explain up to 91.95% of the variation in all variables. Finally, performance metrics demonstrated that Bayesian regularized neural networks achieved the best results, giving an accuracy of $R^2 = 99.99\%$ and RMSE = 0.002 kW.

**Keywords:** artificial intelligence; machine learning; solar energy; forecasting; photovoltaic power

## 1. Introduction

Artificial intelligence is already proving to be a huge success in various sectors and disciplines. Despite the fact that this concept has been there since the 1960s, it has only lately acquired popularity as a result of expanding data quantities, advanced algorithms, and improvements in computing capacity [1].

In the energy field, artificial intelligence can be used as a forecasting tool for grid quality and stability, planning, dispatching of power, and efficient management [2]. Renewable energy sources encounter several critical challenges regarding their integration in the energy mix due to their unpredictability and improbability. In the case of photovoltaic solar energy, these inaccuracies are mainly controlled by the Earth's motion around the sun [3].

The relevance of this problem has led to advanced research in order to accurately predict photovoltaic power production. One of the best solutions used to tackle and address this issue is the machine learning approach since it does not require any knowledge about PV systems. In the literature, several machine learning-based prediction techniques are used, including multiple linear regression (MLR) [4], support vector regression (SVR) [5], random forest (RF) [6], quantile random forest (QRF) [7], long short-term memory (LSTM) neural networks (NNs) [8], K nearest neighbors (KNN), extreme learning machine (ELM), generalized regression neural network (GRNN) [9], elastic net, ridge regression, gradient boosting (GB) [10] etc.

This study can give important information regarding forecasting methodologies to academics and engineers working in solar PV plants, since it presents comparative research

on different machine learning techniques for hourly PV power prediction. Moreover, since several factors, namely climatic variables, can affect solar PV output power and add complexity to the prediction process, a principal component analysis (PCA) was conducted to decrease the number of interconnected variables into a smaller number of dominating factors. The prevailing factors were then used as inputs for the predictive models. Finally, the accuracy of the proposed models was assessed using performance metrics, residual analysis, and a diagnostic approach, mainly the regression error characteristic (REC) curve. The main contributions of this study are the following:

- This study enhances the ability of short-term PV power predictions thanks to the robust and competitive results obtained in terms of $R^2$ and RMSE
- Our approach requires only open data freely available on the web, and anyone with technological skills may create their own customized version.
- The most relevant variables to PV power prediction are identified using PCA.
- Finally, investments in new PV installations will be encouraged thanks to the results of our comparison.

## 2. Materials and Methods

### 2.1. Data Source and Description

In this work, we used the hourly PV output power data (PAC) derived from a PV power platform with a total capacity of 6 kW in Rabat, Morocco. For the input data, we used SoDa, a free data source offering solar energy and weather services. The inputs utilized in our forecasting models are presented in Tables 1–3 as follows.

**Table 1.** Solar irradiation parameters.

| Parameter | Unit | Symbol |
|---|---|---|
| Top of atmosphere radiation | $Wh/m^2$ | TOA |
| Clear sky global horizontal irradiation | $Wh/m^2$ | CSGHI |
| Clear sky beam horizontal irradiation | $Wh/m^2$ | CSBHI |
| Clear sky diffuse horizontal irradiation | $Wh/m^2$ | CSDHI |
| Clear sky beam normal irradiation | $Wh/m^2$ | CSBNI |
| Global horizontal irradiation | $Wh/m^2$ | GHI |
| Beam horizontal irradiation | $Wh/m^2$ | BHI |
| Diffuse horizontal irradiation | $Wh/m^2$ | DHI |
| Beam normal irradiation | $Wh/m^2$ | BNI |
| Short-wave irradiation | $Wh/m^2$ | Irr |

**Table 2.** Meteorological parameters.

| Parameter | Unit | Symbol |
|---|---|---|
| Relative humidity | % | RH |
| Wind speed | $m/s$ | WS |
| Wind direction | deg | WD |
| Ambient temperature | °C | Tamb |
| Pressure | hPa | P |

**Table 3.** Two additional parameters.

| Parameter | Unit | Symbol |
|---|---|---|
| Cell temperature | °C | Tcell |
| Efficiency | % | Eff |

### 2.2. Principal Component Analysis

Principal component analysis (PCA) is an extremely powerful tool for synthesizing information. It is used especially when there is a large amount of quantitative data to

process and interpret. The core of this statistical technique is to use fewer independent factors to reflect the majority of the original variables and to eliminate their duplication [11]. The principal components are obtained from the covariance matrix's eigenvalues and eigenvectors [12]. In this study, 17 variables were studied as inputs for our predictive models.

### 2.3. Machine Learning Algortihms

In this paper, four machine learning algorithms were tested and fitted in R (R Core Team, 2018) [13]. The dataset was partitioned into two parts—training and testing sets—according to the Pareto rule of 80% and 20% using the function createDataPartition in the CARET Package in R. We defined the training and tuning settings using the trainControl function. To minimize over-fitting of the training set, we used cross-validation with 10 folds.

#### 2.3.1. Elastic Net Regression

The first algorithm tested in our work was elastic net regression. It adds two penalty terms from both the lasso and ridge methods to regularize regression models (with non-zero coefficients $\beta_i$) as presented in Equation (1) [14].

$$pen\ (\beta)\ =\ \lambda\ (\alpha\ \sum_{i\ =\ 1}^{k} |\beta_i| + (1 - \alpha)\ \sum_{i\ =\ 1}^{k} |\beta_i|^2) \tag{1}$$

where lambda $\lambda$ ($\lambda \geq 0$) is the penalty coefficient.

### 2.4. Support Vector Regression

The second method used in our study was support vector regression. It is one of the most popular algorithms in machine learning. Its main principle is to find an ideal hyperplane in the training data space that represents all of the observations in the dataset. The hyperplane is the line used to forecast the target. The support vectors or data points nearest to the boundary lines might be either within or outside the boundary lines [6]. The hyperplane is then established by any equation, i.e., non-linear or polynomial. In this study, a radial-based kernel function was used [15].

### 2.5. Random Forest Regression

Random forest (RF) is an ensemble-based regression method. In the form of a tree structure, RF displays relationships between features and the target, which allows for easy-to-understand and interpretable results [16]. This method is a decision tree adaptation, in which a model produces predictions based on a succession of base models as stated as in Equation (2) [17].

$$g(x)\ =\ f1(x) + f2(x) + f3(x) + \cdots + fk(x) \tag{2}$$

where each base model is a decision tree and $k$ denote the number of decision trees.

### 2.6. Bayesian Regularized Neural Networks

In this work, we investigated the ability of a neural network trained using the Bayesian regularization technique to forecast PV power, since this method has not seen many applications in the field of solar energy prediction. The Bayesian technique has a variety of practical benefits, including the ability to solve the over-fitting problem which occurs in conventional neural networks [18].

### 2.7. Performance Metrics

The accuracy of PV power (PAC) forecasting models was evaluated considering the following metrics [19,20]:

$$R^2\ =\ 1 - \frac{\sum_{i\ =\ 1}^{n} \left(PAC_i - P\hat{A}C_i\right)^2}{\sum_{i\ =\ 1}^{n} \left(PAC_i - \overline{PAC}\right)^2} \tag{3}$$

$$RMSE = \sqrt{\frac{1}{n}\sum_{i=1}^{n}\left(PAC_i - P\hat{A}C_i\right)^2} \tag{4}$$

$$MAE = \frac{1}{n}\sum_{i=1}^{n}\left|PAC_i - P\hat{A}C_i\right| \tag{5}$$

## 3. Results

### 3.1. Principal Component Analysis: Factor Extraction Results

The principal component analysis (PCA) was performed on the datasets to identify the most important data features for use in training the machine learning models. Table 4 shows the variance distribution of the principal components (PCs) (PC1–PC17). According to the eigenvalues, it appears that the cumulative variance of PC1 to PC6 is 91.95%Therefore, the first six principal components were identified as the main model inputs and were sufficient to develop our predictive models. Moreover, Figure 1 presents the scree plot, which is a line plot of the correlation matrix's eigenvalues, arranged from greatest to smallest.

**Table 4.** Principal components (PCs).

| PCs | Eigenvalues | Variance (%) | Cumulative Variance (%) |
|---|---|---|---|
| 1 | 9.5770 | 56.3356 | 56.3356 |
| 2 | 1.8230 | 10.7240 | 67.0597 |
| 3 | 1.4733 | 8.6669 | 75.7266 |
| 4 | 1.0447 | 6.1458 | 81.8724 |
| 5 | 0.9068 | 5.3345 | 87.2069 |
| 6 | 0.8063 | 4.7434 | 91.9504 |
| 7 | 0.4738 | 2.7875 | 94.7379 |
| 8 | 0.2733 | 1.6078 | 96.3458 |
| 9 | 0.2175 | 1.2795 | 97.6253 |
| 10 | 0.1779 | 1.0468 | 98.67222 |
| 11 | 0.0958 | 0.5638 | 99.23608 |
| 12 | 0.0778 | 0.4579 | 99.69398 |
| 13 | 0.0407 | 0.2396 | 99.93361 |
| 14 | 0.0099 | 0.0583 | 99.99193 |
| 15 | 0.0013 | 0.0080 | 99.99999 |
| 16 | $2.3198 \times 10^{-6}$ | $1.3646 \times 10^{-5}$ | 100 |
| 17 | $3.6497 \times 10^{-8}$ | $2.1469 \times 10^{-7}$ | 100 |

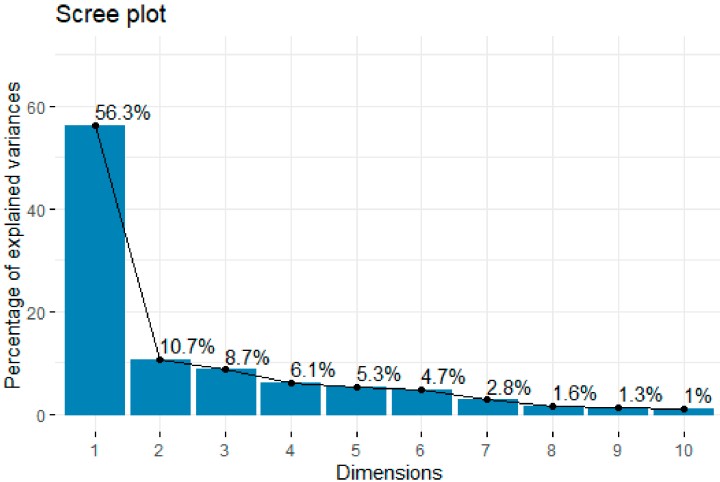

**Figure 1.** Scree plot.

The top three variables with a value greater than 0.60 in Table 5 were chosen as the main variables of each of the PCs to choose prominent predictor variables for further

regression analysis [21]. For PC1, the global horizontal irradiation (GHI), beam horizontal irradiation (BHI), and beam normal irradiation (BNI) were used. For PC2, the pressure (P) was identified. For PC3, top of atmosphere radiation (TOA), clear sky diffuse horizontal irradiation (CSDHI), and diffuse horizontal irradiation (DHI) were chosen. For PC4, the wind speed parameter was selected. For PC5, cell temperature (Tcell), PV efficiency (Eff), and ambient temperature (Tamb) were used. Finally, only wind direction was identified for PC6. All 12 variables were considered to be PV power driving factors, allowing them to be used as inputs in the proposed predictive models.

**Table 5.** Results of PCA.

| Factor | PC1 | PC2 | PC3 | PC4 | PC5 | PC6 |
|--------|-----|-----|-----|-----|-----|-----|
| Tcell | 0.41 | 0.01 | 0.24 | 0.09 | <u>0.83</u> | 0.11 |
| Eff | 0.17 | 0.18 | 0.43 | 0.21 | <u>0.75</u> | 0.04 |
| Tamb | 0.26 | −0.56 | 0.08 | −0.28 | <u>0.62</u> | 0.13 |
| RH | −0.42 | 0.08 | −0.02 | −0.02 | −0.77 | −0.02 |
| P | 0.10 | <u>0.88</u> | −0.11 | −0.24 | 0.04 | −0.08 |
| WS | −0.02 | −0.17 | 0.00 | <u>0.94</u> | 0.10 | 0.08 |
| WD | −0.01 | −0.10 | 0.04 | 0.08 | 0.11 | <u>0.98</u> |
| Irr | 0.75 | −0.12 | 0.33 | 0.04 | 0.46 | 0.13 |
| TOA | 0.73 | −0.06 | <u>0.64</u> | 0.01 | 0.21 | 0.02 |
| CSGHI | 0.77 | −0.04 | 0.59 | 0.02 | 0.19 | 0.02 |
| CSBHI | 0.84 | 0.00 | 0.46 | 0.04 | 0.18 | 0.02 |
| CSDHI | 0.34 | −0.16 | <u>0.84</u> | −0.05 | 0.19 | 0.02 |
| CSBNI | 0.75 | 0.22 | 0.44 | 0.12 | 0.25 | −0.04 |
| GHI | <u>0.89</u> | −0.04 | 0.36 | −0.06 | 0.24 | 0.01 |
| BHI | <u>0.94</u> | −0.04 | 0.12 | −0.08 | 0.24 | 0.00 |
| DHI | 0.29 | −0.05 | <u>0.91</u> | 0.02 | 0.13 | 0.04 |
| BNI | <u>0.87</u> | 0.12 | 0.01 | −0.06 | 0.34 | −0.06 |

Underlined variables are the top three variables with a value greater than 0.60.

### 3.2. Final Models

3.2.1. Elastic Net Regression

The final values used for the model were $\alpha = 0.9$ and $\lambda = 1.884535$. The regression coefficients of the final model are presented below in Table 6.

**Table 6.** Regression coefficients of the final model.

| Input Parameter | Coefficient |
|-----------------|-------------|
| Intercept | $-7.7668 \times 10^3$ |
| Tcell | $8.2463 \times 10^1$ |
| Eff | $-3.3805 \times 10^1$ |
| Tamb | $-9.8509 \times 10^1$ |
| P | $8.9286 \times 10^0$ |
| WS | $7.8149 \times 10^0$ |
| WD | $-7.2022 \times 10^{-1}$ |
| TOA | $7.8705 \times 10^{-2}$ |
| CSDHI | $2.3547 \times 10^{-2}$ |
| GHI | $4.4554 \times 10^{-2}$ |
| DHI | $2.5494 \times 10^{-1}$ |
| BNI | $2.0092 \times 10^{-1}$ |

3.2.2. Support Vector Regression

We used a radial-based kernel function to conduct an epsilon regression. The final model's parameters achieving the best fit were $Cost = 1$, $\gamma = 0.0833$, $\varepsilon = 0.1$, and $number\ of\ support\ vectors = 3641$.

### 3.2.3. Random Forest

In the final model, the number of variables randomly test at each split was $mtry = 7$, as seen in Figure 2.

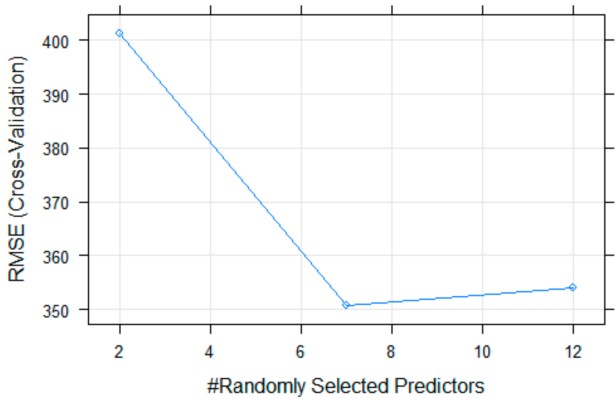

**Figure 2.** Random forest model plot.

### 3.2.4. Bayesian Regularized Neural Networks

As shown in Figure 3, the final value of neurons reducing the RMSE error in the final model was $neurons = 3$.

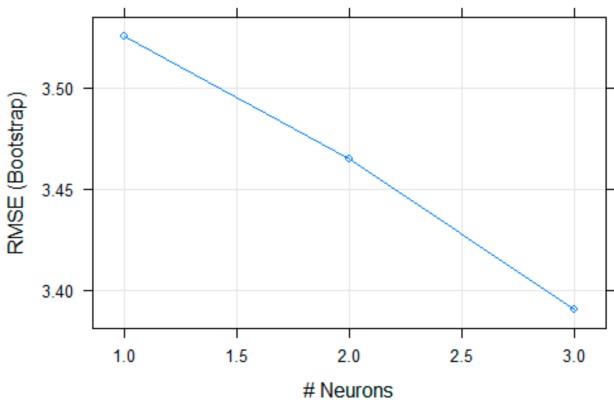

**Figure 3.** Bayesian regularized neural networks model plot.

### 3.3. Performance Measures for Predictive Models

The accuracy of the investigated models was measured for the training phase and the testing phase using the most common metrics in regression as presented in Tables 7 and 8.

**Table 7.** Performance metrics—training phase (80%).

| Machine Learning Algorithm | $R^2$ | RMSE (kW) | MAE (kW) |
|---|---|---|---|
| Elastic net regression | 0.8933 | 0.6900 | 0.5167 |
| Support vector regression | 0.9436 | 0.5014 | 0.3295 |
| Random forest regression | 0.9953 | 0.1434 | 0.0906 |
| Bayesian regularized neural networks | 0.9999 | 0.0033 | 0.0024 |

**Table 8.** Performance metrics—testing phase (20%).

| Machine Learning Algorithm | $R^2$ | RMSE (kW) | MAE (kW) |
|---|---|---|---|
| Elastic net regression | 0.8930 | 0.6957 | 0.5297 |
| Support vector regression | 0.9368 | 0.5344 | 0.3632 |
| Random forest regression | 0.9733 | 0.3489 | 0.2297 |
| Bayesian regularized neural networks | 0.9999 | 0.0027 | 0.0021 |

Scatterplots (see Figure 4) revealed more information about the model's effectiveness. Figure 4 shows the scatterplots of predicted values vs. actual ones. For a suitable model, all points should be near to the diagonal line and show no practical dependencies.

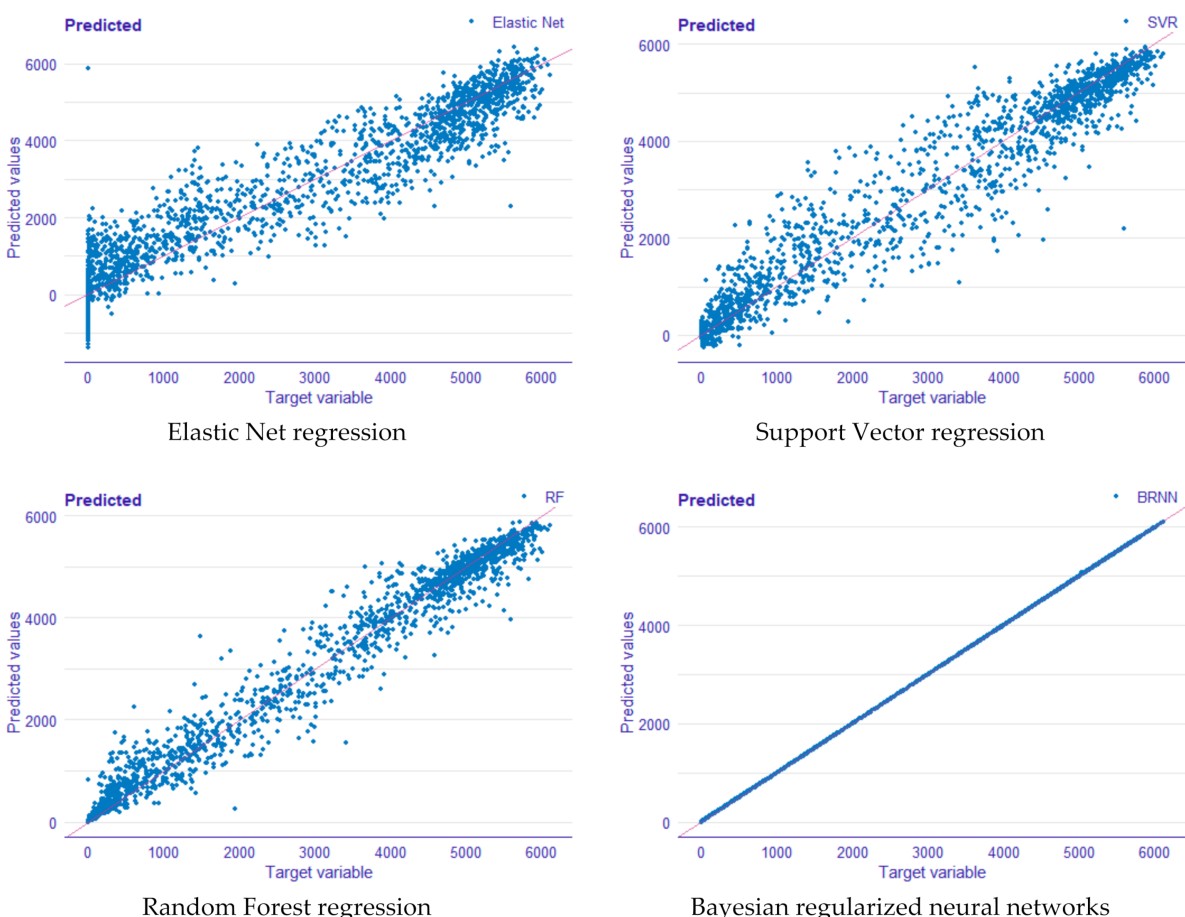

Elastic Net regression

Support Vector regression

Random Forest regression

Bayesian regularized neural networks

**Figure 4.** Scatterplots: predicted versus observed values plots.

### 3.4. Residual Analysis Result

The investigation of residuals is widely acknowledged as a critical step in any regression study. The first plot (see Figure 5) displays the residuals versus the observed values.

Elastic Net regression

Support Vector regression

Random Forest regression

Bayesian regularized neural networks

**Figure 5.** Residuals versus observed values plots.

The second plot represents the residual boxplot. It depicts the distribution of absolute residual values as illustrated in Figure 6.

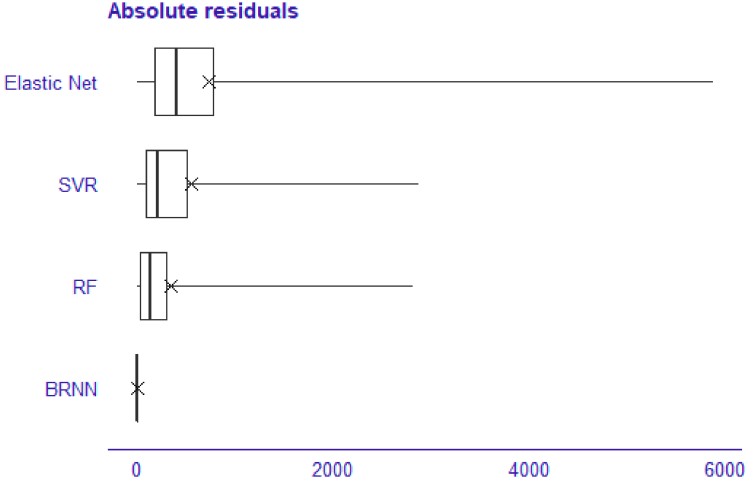

**Figure 6.** Residual boxplots.

The last plot represents regression error characteristic (REC) curve. This is a regression form of the ROC curve in classification. The error tolerance is plotted on the x-axis, and the percentage of points forecasted inside the tolerance is plotted on the y-axis [22].

## 4. Discussion

The principal component analysis (PCA) conducted above revealed six major factor components affecting PV power and explaining up to 90% of the total variable variance. The most significant variables identified using the PCA technique were subsequently used in the proposed models.

Moreover, based on the findings of the performance metrics acquired in Tables 7 and 8, it can be seen that non-linear models, particularly Bayesian regularized neural networks and random forest, obtained the best compromise between the predicted and observed values, with $R^2 = 99.99\%$ and $R^2 = 99.53\%$, respectively, in the training phase and $R^2 = 99.99\%$ and $R^2 = 97.33\%$, respectively, in the testing phase, while the lowest performance was achieved by linear models such as the elastic net algorithm with $R^2 = 89.3\%$ and RMSE = 0.69 kW. This is mainly because non-linear methods are better at including data dynamics and capturing non-linear correlations between variables.

Finally, several plots have been presented above to enable a more accurate study of the models in terms of residuals. For instance, using residual versus observed values plots in Figure 5 showed that Bayesian regularized neural networks offer better prediction accuracy when compared with the other predictive models investigated in this study, since the residuals are symmetrically distributed around the x-axis (near to zero). In the same way, when examining the residual boxplots (see Figure 6), we can observe that Bayesian regularized neural networks have the fewest residuals, followed by random forest and support vector regression, unlike elastic net, which has considerably more widely distributed residuals.

Another important tool for comparing and analyzing the accuracy of regression models for different tolerance levels is the REC curve graphic (see Figure 7). The ideal model is located in the upper left corner, similar to the ROC curve. The better the model, the faster the curve approaches this point, which is the case for the Bayesian regularized neural networks model, followed by random forest.

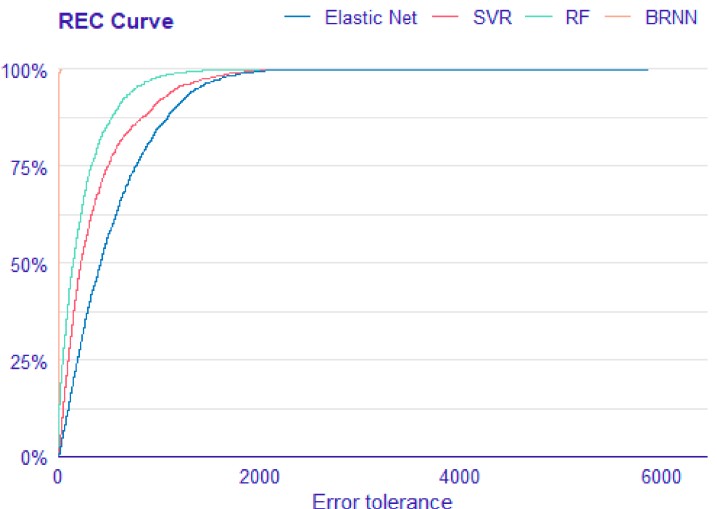

**Figure 7.** REC curves.

This study presented deep insight into comparing the performance of four statistical and machine learning techniques for hourly PV power forecasts, which will be useful to researchers and engineers working in the field of solar photovoltaic energy such as PV-

integrated smart buildings, efficient energy management system, electric vehicle charging, and smart grids.

## 5. Conclusions

Two key contributions are made by this study. To begin, the most significant variables affecting PV power were identified using principal component analysis (PCA). In addition, a comparison research was conducted to explore which algorithms forecasted solar PV output power the best.

PV power prediction will not only aid in assuring cost-effective solar power dispatch, but it will also enable solar electricity suppliers to make better financial and funding decisions. Finally, the presented findings show that machine learning algorithms can accurately forecast the output power generated by PV panels in a shorter amount of time. This specificity is determined by the precision of the data utilized, the time horizon, meteorological conditions, and the geographic area.

**Author Contributions:** Conceptualization; methodology; software; writing—review and editing, S.C.; Supervision, M.M. All authors have read and agreed to the published version of the manuscript.

**Funding:** This research received no external funding.

**Conflicts of Interest:** The authors declare no conflict of interest.

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
