# Peer review of "Principal Component Analysis and Machine Learning Approaches for Photovoltaic Power Prediction: A Comparative Study"

_applsci, doi:10.3390/app11177943_

Round 1
Reviewer 1 Report
The paper explain the use of PCA to reduce the variables to predict the generated power by a Photovoltaic system. The authors explain very well their research, it is well structured and well written.
I only has a few comments, to try to improve the quality of the paper:
- I don't understand understand why the authors choose 18 principal components; I suppose that the testing variables should be only the inputs variables, while PAC is the output of the model. I must say that I don't use PCA in my researches, and I don't know if it is normal or not.
- Figure 3 shows the results of the bayesian regularized neural network, but the lowest error is achieved with the maximum number of neurons; I don't know if the authors tested the model with more neurons to look for a minimum error.
- I don't know how the authors divided the data for train and test: hold-out with 70% 30%? K-fold with 10 folds? They should explain this, and the use of k-fold use to provided more general results.
- I suggest to change the order of plots in Figure 6; all the explanation and the results in the rest of paper follow the same order, but not in that figure.
- In the discussion section, the R2 values are not the same as in Table 7; the best two models don't have the same value (acording to the Table 7), but in the discussion appear the same value. Moreover, the authors should check the decimal "coma", as in the discussion appear two values as "99,99" and "97,3" instead of "99.99" and "97.3".
Reviewer 2 Report
The authors apply four statistical and machine learning methods to a dataset to demonstrate their usefulness. It is not clear how this advances science in the field. This needs to be explained in the introduction and discussion. Additionally, below are some comments regarding sentence flow. Similar issues exist in the body of the manuscript.
- “Nowadays, in the context of the industrial revolution 4.0, considerable volumes of data are being generated continuously from intelligent sensors and connected objects. Thus, the proper understanding and use of these amounts of data are crucial levers of performance and innovation.” The latter sentence doesn’t logically follow from the former. Reconsider the use of “Thus”.
- “In fact, machine learning is the technology that allows the full potential of big data sets to be exploited.” The sentence doesn’t logically follow from the prior. Reconsider the use of “In fact”.
- “In the energy field, the advantages of machine learning forecasts are not merely financial for energy producers, but they also lend legitimacy to renewable energy sources such as solar energies, making their integration into the energy mix easier.” Not logically framed and doesn’t logically follow from previous sentences.
- “Hence, the key goal of this paper is to use machine learning approaches to forecast the hourly power produced by photovoltaic panels.” It’s not clear why “Hence”. Lacking logical chain in sentences.
- “In fact, a comparison analysis of various predictive models including elastic net, support vector regression, random forest and Bayesian regularized neural networks is carried out to identify the models providing the best predicting results.” Not clear why “In fact”.
- “Furthermore, the principal components analysis revealed six main factor components that can explain up to 91.44 % of the variation in all variables.” Not clear why “Furthermore.”
